# Ray–Wave Correspondence in Microstar Cavities

**DOI:** 10.3390/e24111614

**Published:** 2022-11-05

**Authors:** Julius Kullig, Jan Wiersig

**Affiliations:** Institut für Physik, Otto-von-Guericke-Universität Magdeburg, Postfach 4120, D-39016 Magdeburg, Germany

**Keywords:** microcavities, quantum chaos, ray–wave correspondence

## Abstract

In a previous work published by the authors in 2020, a novel concept of light confinement in a microcavity was introduced which is based on successive perfect transmissions at Brewster’s angle. Hence, a new class of open billiards was designed with star-shaped microcavities where rays propagate on orbits that leave and re-enter the cavity. In this article, we investigate the ray–wave correspondence in microstar cavities. An unintuitive difference between clockwise and counterclockwise propagation is revealed which is traced back to nonlinear resonance chains in phase space.

## 1. Introduction

Optical microcavities are fascinating systems to confine light on a very small scale [1,2]. Traditional whispering-gallery cavities rely on successive total internal reflections at the cavity’s dielectric interface which prevent a ray from leaving the cavity. Correspondingly, long-lived whispering-gallery modes exist as solutions of Maxwell’s equations. Such whispering-gallery cavities appear, e.g., as microdisks [3,4,5,6], microtoroid [7,8,9], microspheres [10], or microjet cavities [11]. They have attracted an immense amount of research interest, e.g., for quantum chaos [12,13,14,15] or exceptional points in non-Hermitian systems [16,17,18,19,20] and, additionally, offer a lot of applications, e.g., the generation of optical frequency combs [21], as microlasers with directional emission [22,23,24,25], optical sensors for nanoparticles [26,27,28] or rotating motion [29], or orbital angular momentum lasers [30].

In a former article [31], an alternative concept for light confinement was introduced which is completely different from the traditional whispering-gallery design. The perfect transmissions at Brewster’s angle are utilized such that a ray can leave (and reenter) the cavity without partial back reflection. Thus, via successive transmissions through the dielectric interface of the cavity, rays are guided along a periodic orbit without loss of intensity. This idea can be implemented in a star-shaped cavity as shown in Figure 1a or in Brewster-notched cavities [32].

In this article, we further investigate the ray–wave correspondence in the microstar cavity. In particular we look at (asymmetric) deformed microstar cavities where the formation of the modes is governed by the regular dynamics around an elliptical fixed point in phase space. In the semiclassical regime, we observe modes localizing along the regular structures in phase space, while this localization might be expected from quantum-chaos theory, we can also identify an unexpected difference between clockwise (CW) and counterclockwise (CCW) propagation in the star cavity. In the previous literature, a general imbalance of CW and CCW propagation due to asymmetric backscattering was discussed for microcavities [19,33,34,35]. However, in this article, we study the difference pattern between CW and CCW propagation that is due to a nonuniform loss across a resonance chain. Therefore, the revealed mechanism is a general feature of open systems and not restricted to asymmetric boundary deformations. In addition, a Frobenius–Perron operator (FPO) formalism is used as a connection between ray dynamics and optical modes.

The paper is organized as follows. In Section 2, the ray dynamics of the microstar cavity are issued with a special focus on the dynamics of CW and CCW propagation in phase space. The corresponding impacts on the wave dynamics are discussed in Section 3. A conclusion is given in Section 4.

## 2. Ray Dynamics and Phase-Space Description for Microstar Cavities

A microstar is a quasi-two-dimensional cavity with homogeneous refractive index *n* [31]. It is characterized by the number of spikes ν. In order to guide a ray via successive transmissions at Brewster’s angle $\chi_{B}=\arctan\left(1/n\right)$ along a closed orbit, both the refractive index and the opening angle α of the spikes need to be adjusted as
(1)$$\begin{array}{ccc} n & = & \frac{1+\sin(\pi /\nu )}{\cos(\pi /\nu )} \end{array}$$
(2)$$\begin{array}{ccc} \alpha & = & \frac{\nu -2}{2\nu }\pi . \end{array}$$In the following, a microstar with ν=9 spikes is considered with a refractive index of n≈1.43 and spike opening angle $\alpha =70^{\circ }$. The microstar with such a polygonal shape supports a family of periodic orbits propagating without loss of intensity. However, these periodic orbits are only marginally stable. A slight boundary deformation of the microstar can be used to stabilize the dynamics around one central orbit as shown in Figure 1a. In polar coordinates (r,ϕ), the boundary of the most right deformed spike can be expressed as
(3)$$\begin{array}{ccc} r(\phi ) & =\frac{1}{n\sin(\phi )+\cos(\phi )}-\varepsilon_{1}\phi \left(\phi -\frac{\pi }{\nu }\right) & \;\mathrm{for}\;\phi \in [0,\pi /\nu ] \end{array}$$
(4)$$\begin{array}{ccc} r(\phi ) & =\frac{1}{n\sin(\phi )+\cos(\phi )}+\varepsilon_{2}\phi \left(\phi +\frac{\pi }{\nu }\right) & \;\mathrm{for}\;\phi \in [-\pi /\nu ,0] \end{array}$$
with $\epsilon_{1}$ ($\epsilon_{2}$) being the deformation parameter of the upper (lower) segment of the spike. Thus, the boundary of the deformed microstar is given by periodic continuation. Choosing $\varepsilon_{1}\neq \varepsilon_{2}$ allows for an asymmetric boundary deformation without a mirror reflection symmetry of the cavity.

The ray dynamics in the microstar are conveniently expressed in a Poincaré phase-space section, in the following phase space for short, which is introduced in Figure 1. Whenever a ray propagating along the spikes is passing the abscissa, its position x>0 and the transversal momentum p=sinθ are recorded. For a microstar cavity without deformed boundary, i.e., $\epsilon_{1}=\epsilon_{2}=0$, a family of marginally stable orbits exists at p=0, whereas in the deformed case, an elliptical fixed point with a region of regular dynamics can be achieved. For a boundary deformation with a mirror-reflection symmetry ($\epsilon_{1}=\epsilon_{2}$), the fixed point is either at p=0 or occurs pairwise at ±|p|. Contrary, for an asymmetric deformation ($\epsilon_{1}\neq \epsilon_{2}$), a single elliptical fixed point can occur at a finite nonzero momentum, as in Figure 1c with $\left(\epsilon_{1},\;\epsilon_{2}\right)=\left(0.76,1.2\right)$. For this deformation, a 5:1 resonance chain exists in the regular region, which is of further interest below. In comparison, the deformation parameters $\left(\epsilon_{1},\;\epsilon_{2}\right)=\left(0.78,1.212\right)$ lead to an island of regular dynamics with a similar shape but without a (macroscopic) resonance chain, see Figure 1d.

Note that the phase space represents CCW propagating rays. The corresponding rays propagating in CW direction are given by time reversal of the (geometrical) dynamics. For the phase-space variable *p* describing the momentum along the *x*-axis, this relates to a change in the sign, cf. Figure 1b. A mirror reflection, however, would not change the sign of *p*. Therefore, even in a cavity with a mirror-reflection symmetry, CW and CCW dynamics can have differences, namely in the direction along the symmetry line.

**Figure 1 entropy-24-01614-f001:**
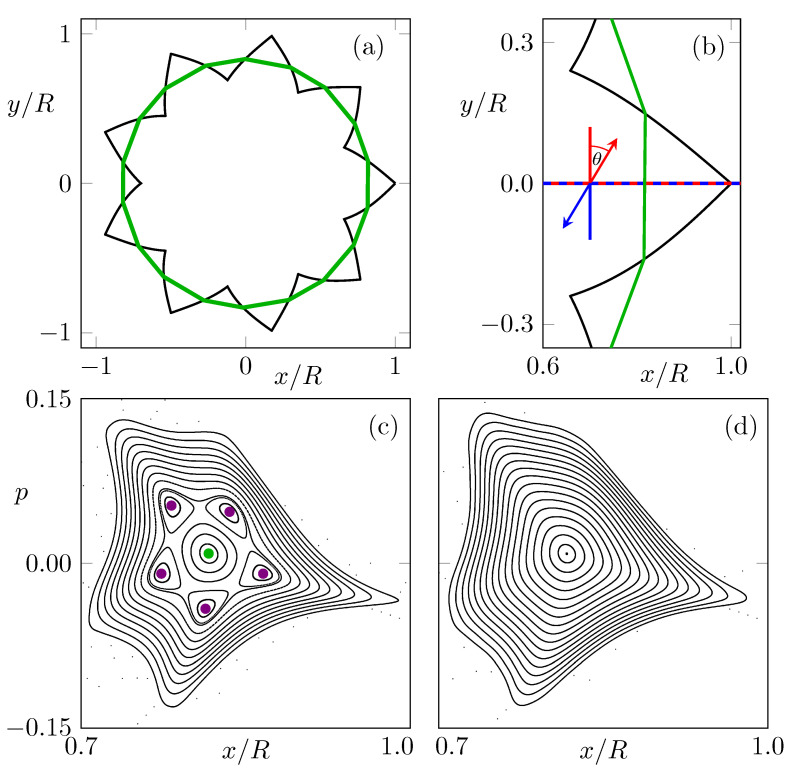
In (**a**), a microstar cavity with ν=9 asymmetric spikes is shown as a black curve. The simple periodic orbit is shown by the green curve. In (**b**), a magnification of one spike is shown. Propagation in CCW (CW) direction is illustrated by red (blue) arrows. Figure (**c**) shows a phase-space portrait of the microstar with deformation parameters $\left(\epsilon_{1},\;\epsilon_{2}\right)=\left(0.76,1.2\right)$, as in (**a**,**b**). The simple periodic orbit is shown as a green dot. A period-5 orbit is shown as magenta dots. (**d**) Phase-space portrait for deformation parameters $\left(\epsilon_{1},\;\epsilon_{2}\right)=\left(0.78,1.212\right)$.

In the following, we analyze the difference between CW and CCW propagation. Therefore, initial conditions $\left(x_{0},p_{0}\right)$ on a uniform grid in phase space with unit intensity $I_{0}=1$ are iterated $N_{\mathrm{turns}}$ turns in CW and in CCW direction. The remaining fractions of the intensity $I_{\mathrm{CCW}}$ in CCW direction are shown in Figure 2a–d for $N_{\mathrm{turns}}=1,15,50,100$, respectively. As can be seen, the loss introduced by partial back-reflections from Fresnel formulas is not uniform in the regular region, while the central fixed point suffers almost no loss due to Brewster-angle transmissions, the border of the regular region has more loss, which results in a loss gradient over the resonance chain. A consequence is a distortion of the intensity pattern, which can be seen in Figure 2b,c. This distortion is different for both propagation directions, resulting in a remarkable difference $I_{\mathrm{CCW}}-I_{\mathrm{CW}}$ around the resonance chain; see Figure 2e–h. Note that the effect becomes considerably weaker for long times where the only remaining intensity is centered around the central fixed point.

Intuitively, the difference between both propagation directions can be understood by analyzing the dynamics around the resonance chain, as shown in Figure 3. Focusing on one unstable orbit of the resonance chain (see red dot in Figure 3), the dynamics on the left side push the orbits closer to the center of the regular region, whereas on the right side, the orbits are pushed away from the regular region’s center. Hence, orbits on the left side of the unstable orbit will suffer less loss then orbits on the right side in the next iterations. This behavior is inverted when the dynamics are reversed from CCW to CW, which leads to the characteristic pattern of the intensity difference in Figure 2f–h.

For comparison, in Figure 4, the system without (macroscopic) resonance chain is investigated. Note that here the difference between intensities $I_{\mathrm{CCW}}$ and $I_{\mathrm{CW}}$ is much weaker and not structured in phase space.

### Frobenius–Perron Operator

The Frobenius–Perron operator (FPO) describes the evolution of density (intensity) distributions in phase space [36]. It has been used for, e.g., maps [37,38,39,40,41], billiard systems [37], and the asymmetric backscattering in deformed microcavities [42]. Here, the aim is to construct classical intensity eigenstates for CCW and CW propagation in the deformed microstar via the FPO. For this purpose, Ulam’s method [43] is used to construct a matrix approximation F of the FPO as follows. The part of the phase space between x∈[r(ϕ/ν),r(0)] and p∈[−0.2,0.2] covering the regular island is divided into a lattice of N×N uniform rectangular cells. In the cell *i*, random initial conditions with total intensity $I_{i}$ are iterated once in (C)CW direction. The transferred intensity $I_{f}$ to the phase-space cell *f* is then recorded. Thus, the two $N^{2}\times N^{2}$ matrices with the elements $F_{fi}^{(C)CW}=I_{f}/I_{i}$ are an approximation for the intensity transport in CW and CCW direction, respectively. As such, the eigenvalues are inside the unit circle, as shown in Figure 5a,b. Of special interest are the eigenstates of the FPO with the eigenvalue of largest magnitude, as they represent states with a long lifetime. For the system with a resonance chain, these long-lived eigenstates for (C)CW propagation are shown in Figure 5c–k, in addition to their differences. As it can be seen, the localization of the eigenstates around the central fixed point depends on the fineness of the discretization given by *N*. This is consistent with the previous literature on FPO eigenstates of maps with a mixed phase space [39,44]. For an infinite discretization, the intensity is localized on the stable periodic orbit, which is a single point in phase space. However, for a more rough discretization, i.e., N≲200, the eigenstates extend over the resonance chain leading to a local intensity distortion, which is different for CW and CCW eigenstates. Consequently, the difference between CW and CCW eigenstates is significant around the resonance chain and consistent with the ray dynamics shown in Figure 2f,g. In particular, the pattern with alternating preference of CW or CCW propagation along the stable and unstable points of the resonance chain is visible.

Note that the shown difference pattern between CW and CCW propagation is not due to an asymmetry of the cavity’s boundary. In fact, for deformation parameters $\epsilon_{1}=\epsilon_{2}=0.98$, the microstar has a mirror-reflection symmetry, a similar 5:1 resonance chain in the phase space, and a similar alternating pattern of the eigenstate difference (not shown).

In Figure 6, the eigenstates corresponding to the second, third, and fourth eigenvalues next to unity are shown. As such an eigenstate has positive and negative parts, it does not represent a proper intensity distribution by itself. However, for the dynamics of an arbitrary (positive) phase-space intensity on intermediate timescales, these eigenstates still provide useful information. Especially, for the second (Figure 6a,b) and fourth (Figure 6e,f) eigenstates, the chirality around the resonance chain as well as the difference between CW and CCW propagation can be seen clearly.

For a comparison, the eigenvalues and eigenstates of the FPO for the system without a resonance chain are shown in Figure 7. The eigenstates also localize at the center of the regular region, but they do not show a significant chiral behavior, i.e., the intensity distribution along the regular orbits is uniform.

Note that in contrast to ref. [42], here, the true-time mapping is not included in the FPO. However, this incorporation of different time scales for the iteration is not relevant here, as the rays propagating through the spikes of the microstar have almost the same optical path length regardless of whether they propagate with a large or small radius.

## 3. Wave Dynamics

In this section, the wave dynamics of the microstar are analyzed with a focus on the semiclassical regime and the difference between CW and CCW propagation. For quasi-two-dimensional microcavities, Maxwell’s equations reduce to the scalar mode Equation [45]
(5)$$\left[\Delta +n^{2}k^{2}\right]\psi =0$$
where k=ω/c is the complex frequency which determines the *Q*-factor (lifetime) of a mode via Q=−Rek/(2Imk). To utilize Brewster’s angle, here, transverse electrical polarization is chosen such that the wave function ψ represents the magnetic field as H→=(0,0,Re[ψe−iωt]). Hence, along the cavities interface ψ, and its scaled normal derivative $n^{-2}\partial_{\vec{v}}\psi$ are continuous. In addition, ψ is required to fulfill Sommerfeld’s outgoing wave condition. For the numerical calculation of the long-lived modes in the microstar, the boundary element method [46] and FEM software COMSOL [47] are used. In Figure 8, the long-lived modes in the complex frequency plane in the semiclassical regime with 350≤RekR≤600 are shown. As in the wave regime discussed in ref. [31], the most long-lived mode localizes along the stable periodic orbit passing through the spikes of the microstar; see Figure 9d. However, additional long-lived modes exist in the semiclassical regime that also pass through the spikes but have more intensity maxima in radial direction; see Figure 9a–c. Thus, a classification of these modes with a radial mode number *l* is suitable.

To compare the wave and ray dynamics in phase space, the Husimi function $H(x,p;\psi ,\partial_{\nu }\psi )$ of a mode ψ is used. Here, we adapt the boundary Husimi function [48] to the phase space taken along the *x*-axis (see Figure 1). Thus, for x∈[0.5R,1.1R], the Husimi function (for CCW propagation) is constructed via
(6)$$H\left(x,p;\psi ,\partial_{\nu }\psi \right)=\frac{1}{N}{\left|kFh_{\psi }\left(x,p\right)+\frac{i}{F}h_{\partial_{\nu }\psi }\left(x,p\right)\right|}^{2}$$
where $F=\sqrt{n\sqrt{1-p^{2}}}$ and $\sigma =\sqrt{R/(nk)}$. The functions $h_{\psi }$ and $h_{\partial_{\nu }\psi }$ are defined via an overlap with a Gaussian wave packet with
(7)$$\begin{array}{ccc} h_{f}\left(x,p\right) & = & \int_{x_{min}}^{x_{max}}\zeta \left(t;x,p\right)f\left(t\right)\;dt \end{array}$$
(8)$$\begin{array}{ccc} \zeta (t;x,p) & = & \exp\left(\frac{{\left(x-t\right)}^{2}}{2\sigma^{2}}-ipt\right). \end{array}$$The normalization constant N in Equation (Equation 6) is chosen such that $\max_{x,p}\;H=1$. Note that for the magnetic field, the normal derivative along the *x*-axis can be computed via $\partial_{\nu }\psi =\partial_{y}H_{z}=-i\omega \varepsilon_{0}\varepsilon_{r}E_{x}$.

In Figure 9e–h, the Husimi functions of the corresponding modes are shown in the right panel. As an expected result, the Husimi function of the most long-lived mode localizes in the center of the regular island, see Figure 9h, with a maximum at the finite momentum p≈0.01, which nicely reflects the stable periodic orbit. Furthermore, as an expected result, the modes with higher radial mode numbers *l* have their Husimi functions localized on outer tori of the regular island. Hence, these states can be seen as higher-order quantized states on the regular region. However, in this regime of RekR≈404, the modes cannot localize on the smaller sub-islands of the resonance chain itself. Consequently, the Husimi functions of the related modes interpolate through the resonance chain.

To verify the different intensity distortion for CCW and CW propagation, a few individual modes with a larger dimensionless frequency RekR>1200 are calculated. Then, for a mode, the Husimi function for CCW propagation $H(x,p;\psi ,\partial_{\nu }\psi )$ and the corresponding Husimi function for CW propagation $H(x,-p;\psi ,-\partial_{\nu }\psi )$ are calculated to obtain the difference.
(9)$$I\left(x,p;\psi \right)=H\left(x,p;\psi ,\partial_{\nu }\psi \right)-H\left(x,-p;\psi ,-\partial_{\nu }\psi \right).$$Note that in Equation (Equation 9), each Husimi function is normalized as discussed above. Hence, a preferred sense of rotation in the mode ψ does not impact the difference I(x,p;ψ). In Figure 10a–i, the Husimi function and the difference I(x,p;ψ) are shown for the long-lived modes in the microstar with a resonance chain. It is remarkable that over the range of kR≈1200, kR≈1280 and kR≈1360, each mode and its partner mode with a quasi-degenerate frequency show almost the same pattern in I(x,p;ψ). This pattern is organized around the stable and unstable fixed point of the resonance chain and in a very good agreement with the pattern of the FPO eigenstates, e.g., in Figure 5e.

To highlight the importance of the resonance chain for this phenomenon, a mode pair in the microstar without macroscopic resonance chain is shown in Figure 10j–l. Already within the mode pair, the pattern I(x,p;ψ) is different and not organized around an obvious phase-space structure.

A simplified model system that explains the effect in an open wave (quantum) system is discussed in the Appendix A.

## 4. Conclusions

In this article, we discussed the ray–wave correspondence in microstar cavities. From the ray dynamics, a (asymmetric) boundary deformation is used to manipulate the light around a periodic orbit with a regular island in phase space. Correspondingly, in the semiclassical regime, optical modes localize hierarchically on the regular island. Depending on the deformation of the microstar resonance, chains might appear in the regular island. Since the intensity loss is not uniform in the regular region, those resonance chains have an unusual effect on the dynamics. The propagation in CW and CCW direction shows an intensity distortion with a pattern that organizes along the resonance chain. This distortion is explained by the ray dynamics, and therefore manifests in the eigenstates of the FPO. Furthermore, the difference between CW and CCW propagation is revealed in the Husimi functions of the optical modes in the semiclassical regime.

Such an interplay between nonuniform loss and nonlinear dynamics is a rather general effect, as it can also be observed in a simple quantum model system. Hence, the effect is not restricted to microstar cavities. However, microstar cavities are ideal systems to study such dynamics, as a nontrivial connection between forward and backward dynamics in time manifests naturally in terms of CW and CCW propagation. Therefore, we believe our studies highlight microstar cavities as an interesting class of systems to unveil novel phenomena.

## Figures and Tables

**Figure 2 entropy-24-01614-f002:**
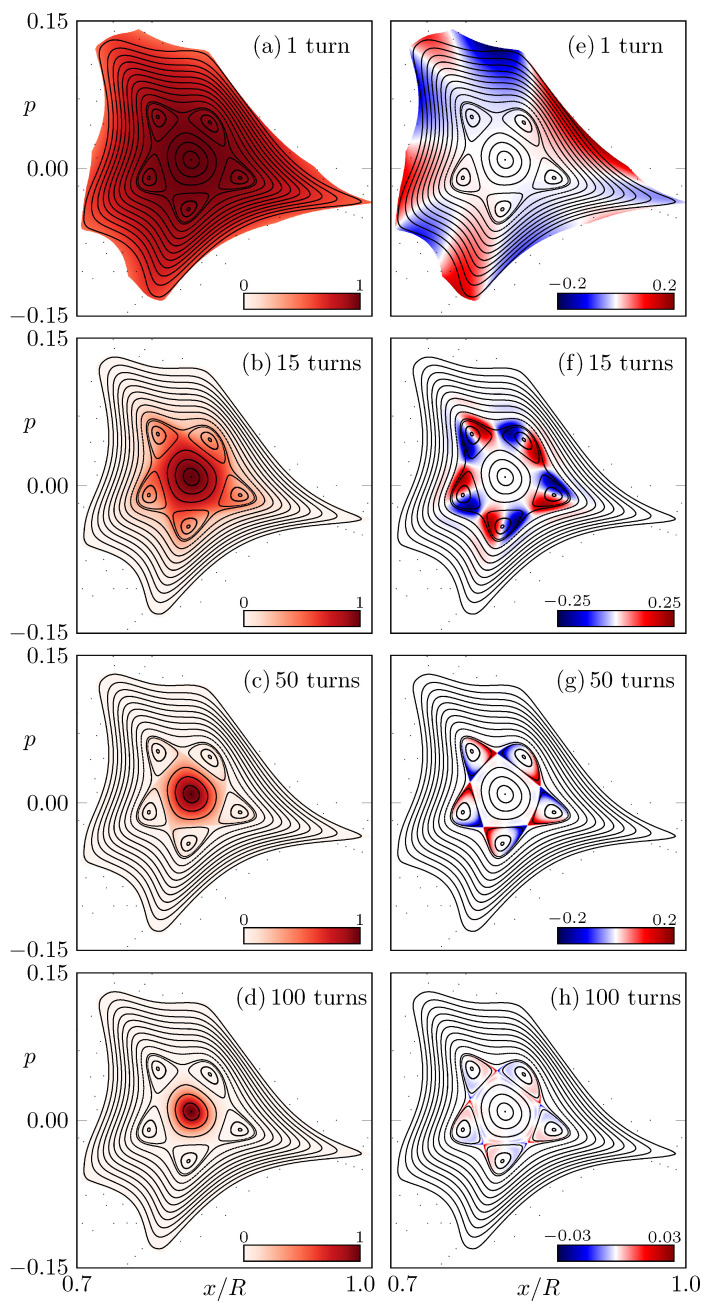
The intensity distribution $\left(x_{0},p_{0},I_{\mathrm{CCW}}\right)$ of rays with initial conditions $\left(x_{0},p_{0},I_{0}=1\right)$ after 1,15,50,100 turns in CCW direction is shown in (**a**–**d**). Correspondingly, (**e**–**h**) show the distribution $\left(x_{0},p_{0},I_{\mathrm{CCW}}-I_{\mathrm{CW}}\right)$ of the difference between propagation in CCW and CW direction. Note the adjusted scale in the colormap for (**e**–**h**).

**Figure 3 entropy-24-01614-f003:**
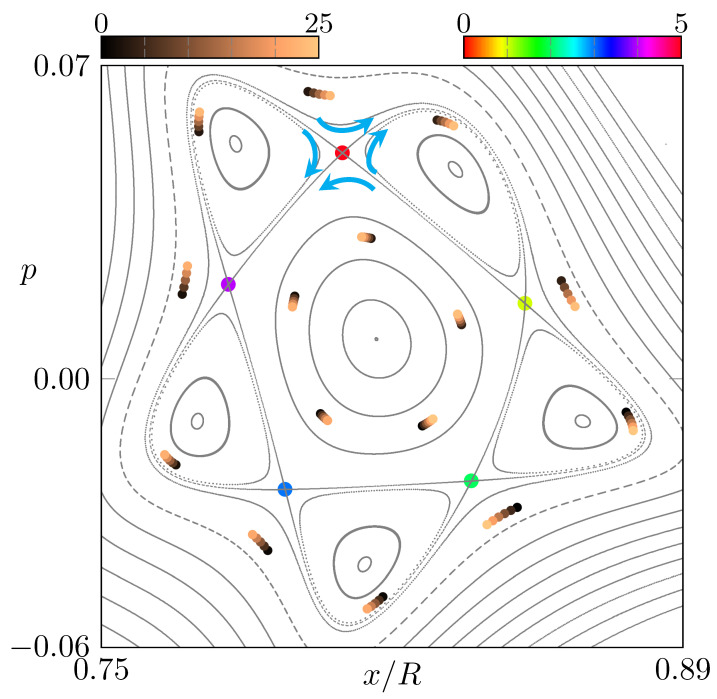
Magnification of the phase-space portrait from Figure 1c around the 5:1 resonance chain. Iterations for selected (nonperiodic) orbits are shown as dots colored from black to orange. The unstable period-5 orbit is represented by dots with a cyclic colormap. The arrows illustrate the dynamics around one of the unstable fixed points. The deformation parameters of the microstar are $\left(\epsilon_{1},\;\epsilon_{2}\right)=\left(0.76,1.2\right)$.

**Figure 4 entropy-24-01614-f004:**
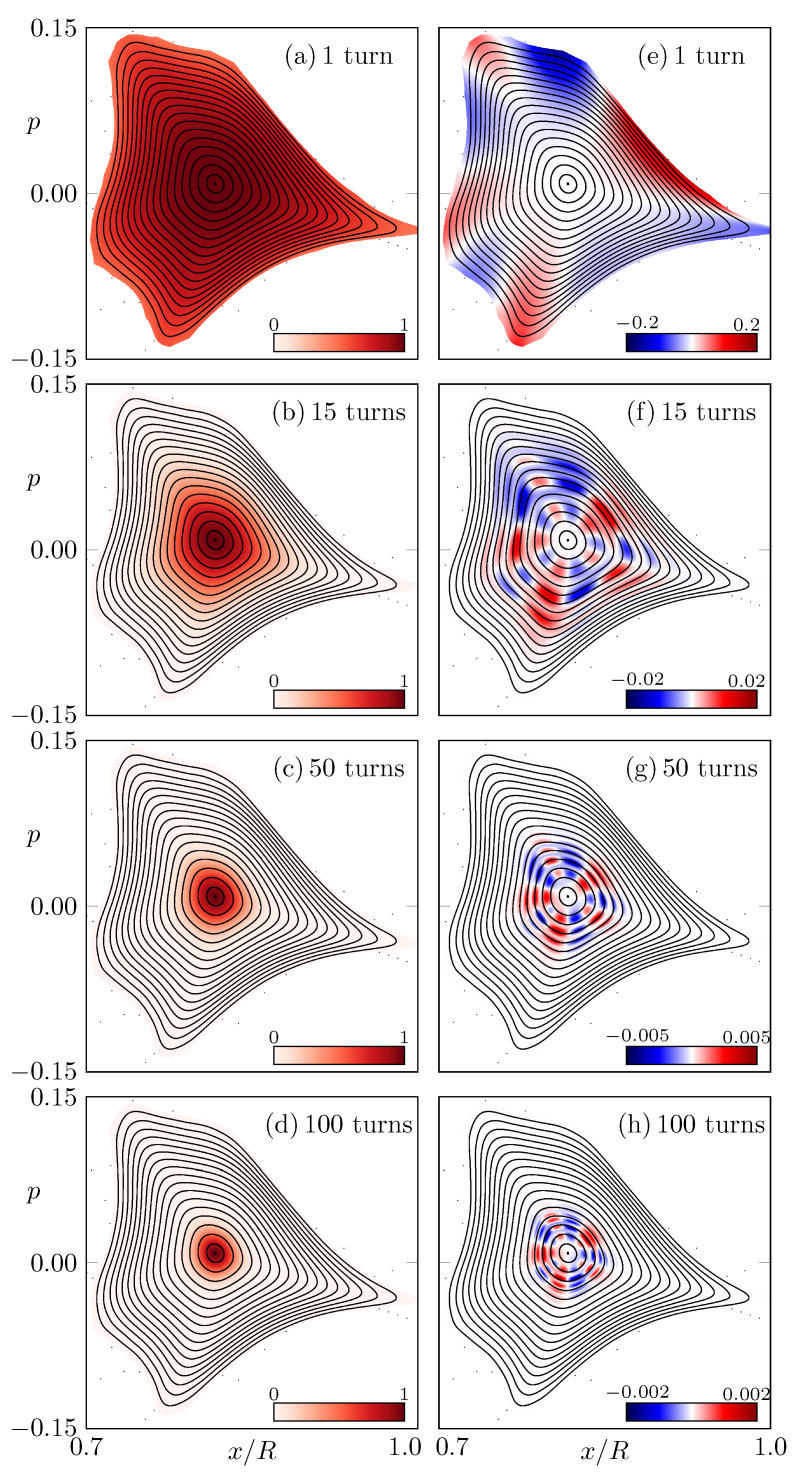
Similar as in Figure 2, (**a**–**d**) show the intensity distributions $\left(x_{0},p_{0},I_{\mathrm{CCW}}\right)$ and (**e**–**h**) show the distributions $\left(x_{0},p_{0},I_{\mathrm{CCW}}-I_{\mathrm{CW}}\right)$, but for deformation parameters $\left(\epsilon_{1},\;\epsilon_{2}\right)=\left(0.78,1.212\right)$, where no macroscopic resonance chain occurs in the phase space. Note the adjusted scale in the colormap for (**e**–**h**).

**Figure 5 entropy-24-01614-f005:**
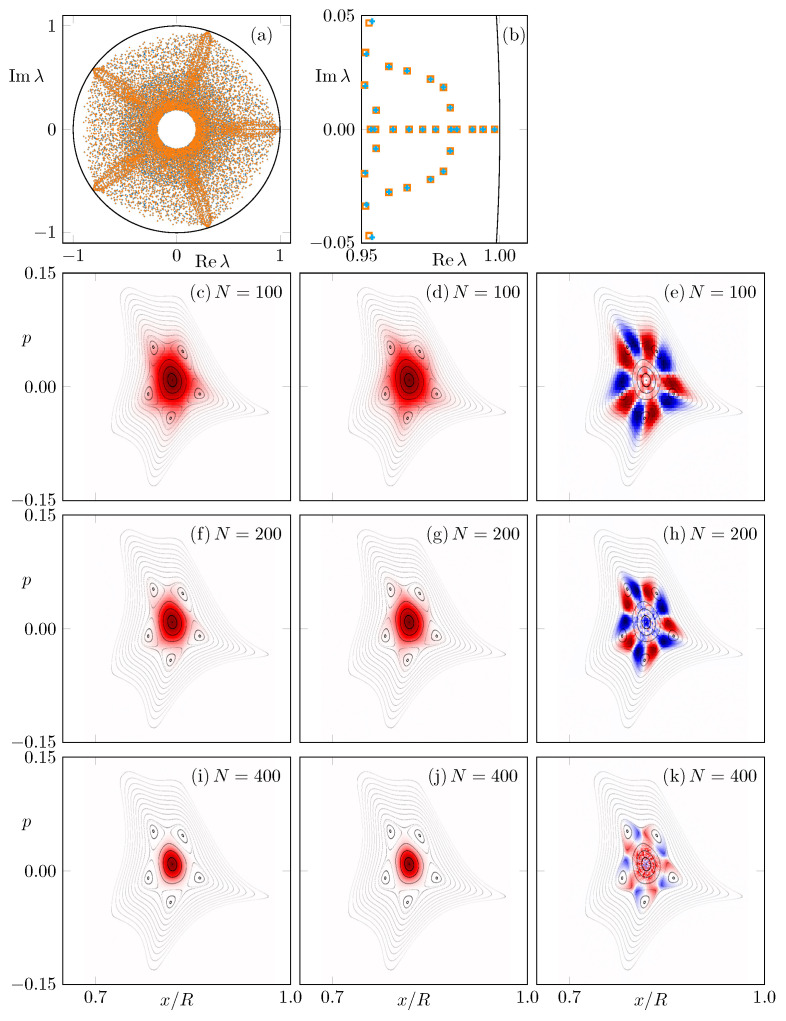
(**a**) The 10,000 largest-magnitude eigenvalues λ of the FPO with N=200 are shown in the complex plane. Orange squares (blue crosses) represent the eigenvalues of the CCW (CW) FPO. Note that they are often on top of each other. (**b**) Shows a magnification of (**a**) around unity. (**c**–**e**) show the eigenstates corresponding to the eigenvalue of the largest modulus for CCW direction, CW direction, and the difference between both eigenstates for N=100. The same sequence is shown in (**f**–**h**) for N=200 and in (**i**–**k**) for N=400.

**Figure 6 entropy-24-01614-f006:**
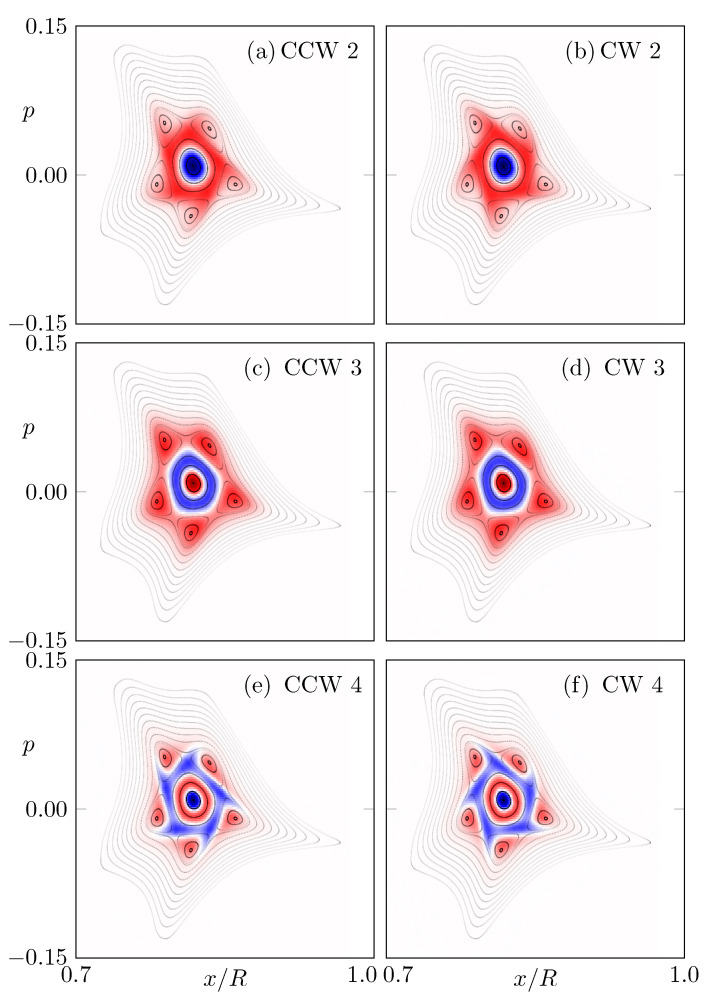
Eigenstates of the FPO (N=200) with (**a**,**b**) second, (**c**,**d**) third, and (**e**,**f**) fourth eigenvalues next to unity. The left (right) panel shows the eigenstates for CCW (CW) propagation.

**Figure 7 entropy-24-01614-f007:**
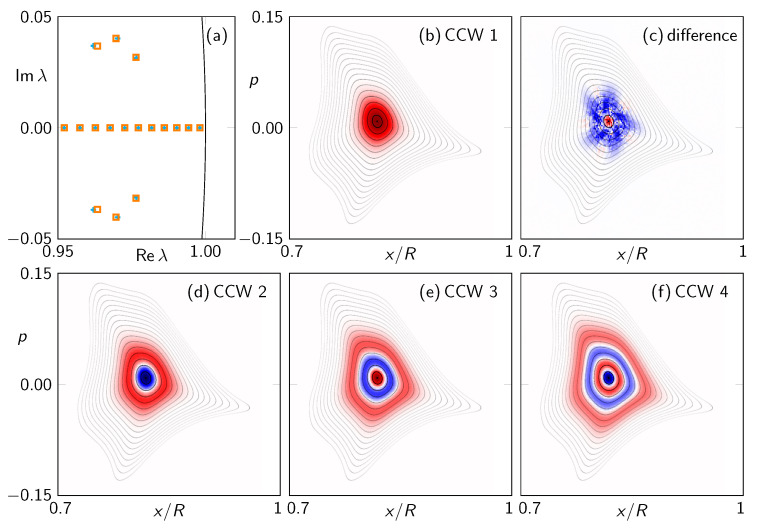
(**a**) Eigenvalues close to unity of the FPO (N=200) for the system without a resonance chain characterized by $\left(\epsilon_{1},\;\epsilon_{2}\right)=\left(0.78,1.212\right)$ (cf. Figure 4). (**b**) Eigenstate of the FPO for the CCW dynamics with largest-modulus eigenvalue. (**c**) Difference between CCW and CW eigenstate. (**d**–**f**) Next eigenstates of the FPO in CCW direction with eigenvalues close to unity.

**Figure 8 entropy-24-01614-f008:**
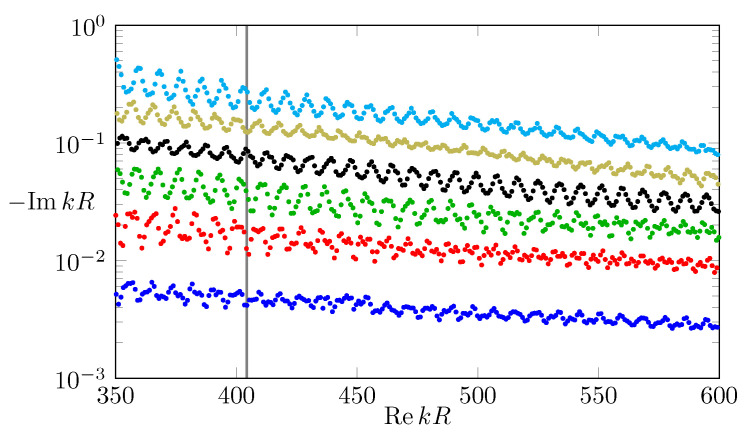
Dimensionless frequencies kR of the long-lived modes in the microstar cavity that can be classified by their mode number in radial direction by (blue) l=1, (red) l=2, (green) l=3, (black) l=4, (yellow) l=5, and (cyan) l=6. The gray line indicates the frequency of the modes shown in Figure 9.

**Figure 9 entropy-24-01614-f009:**
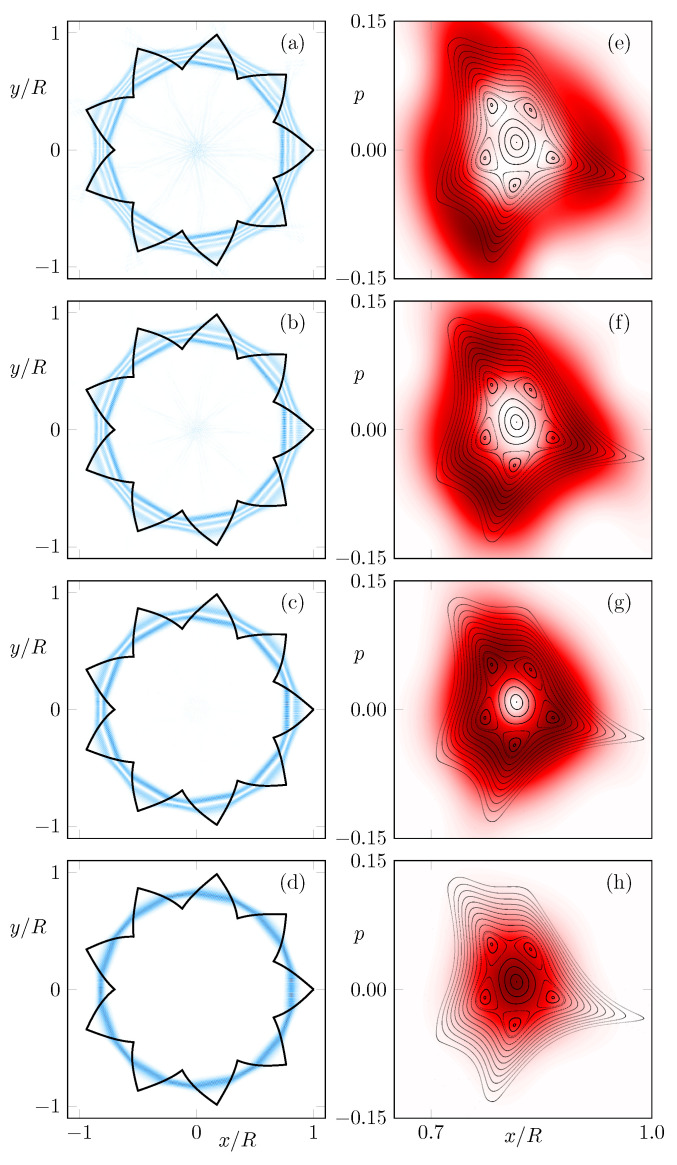
(**a**–**d**) shows the intensity pattern of the modes with radial mode number l=4 to l=1 and RekR≈404 (see gray line in Figure 8). (**e**–**h**) represents the Husimi projection of the corresponding mode.

**Figure 10 entropy-24-01614-f010:**
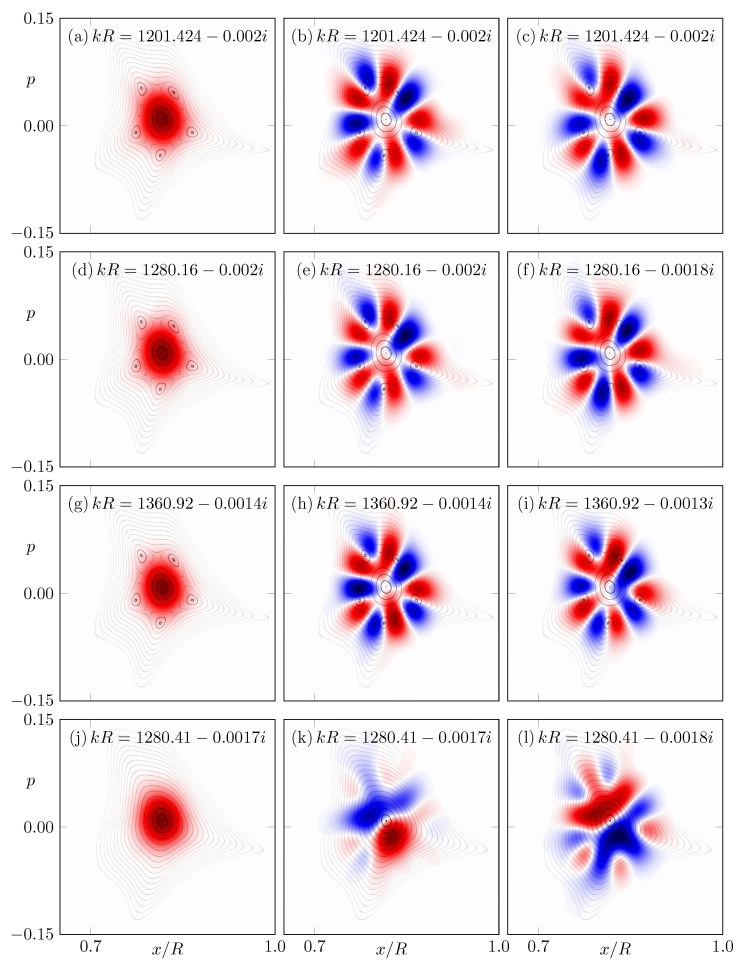
(**a**) Husimi function for CCW propagation, (**b**) difference I(x,p;ψ) between Husimi functions for CCW and CW propagation, see Equation (Equation 9). (**c**) Same as (**b**) but for a partner mode with almost the same frequency kR as in (**b**). The panels (**d**–**i**) follow the same sequence as (**a**,**b**) but for modes with different frequency kR in the same system given by deformation parameters $\left(\epsilon_{1},\;\epsilon_{2}\right)=\left(0.76,1.2\right)$. Modes for the microstar given by $\left(\epsilon_{1},\;\epsilon_{2}\right)=\left(0.78,1.212\right)$ are shown in (**j**–**l**).

## Data Availability

Not applicable.

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
