# Peer review of "Ray–Wave Correspondence in Microstar Cavities"

_entropy, 2022, doi:10.3390/e24111614_

Round 1
Reviewer 1 Report
The authors studied asymmetric microstar cavities where resonance wave functions are localized around stable periodic orbits. The authors found the difference between CW and CCW rotating ray-dynamical trajectories associated with a Poincare-Birkhoff island chain in the phase space. This difference is expressed as a time-invariant object in both ray and wave dynamics., that is, the eigenstate of the Frobenius-Perron operator in the phase-space and the resonance wave function. Through the difference between CW and CCW propagating light, the ray-wave correspondence is clearly shown in the phase space. This manuscript is very relevant to the special issue "Quantum Chaos".
Therefore, I recommend this manuscript for publication in Entropy.
I found the follwing typos:
l15: attract -> attracted
l52: left -> right
l204: belief -> believe
Author Response
We thank the referee for the very positive evaluation of our
manuscript.
The typos have been fixed in the revised version. Figures 5 and
10 have been replaced due to a small correction.
Reviewer 2 Report
In their previous paper, the authors introduced a star-shaped cavity,
which has a special periodic orbit leaving and reentering the
cavity. The cavity shape is tuned so that the reentrance angle is
Brewster's angle, enabling perfect transmission for the periodic
orbit. In this paper, they continue to study ray-dynamical and
wave-dynamical properties of this cavity, especially focusing on the
difference between clockwise (CW) and counter-clockwise (CCW)
propagation. They found that nonlinear resonance chains around the
special periodic orbit affects the difference.
I think the authors should state more clearly and more in detail what
are the main claims of this paper, and how they are important. They
emphasize that their findings are unintuitive and unexpected. However,
given the asymmetry of the cavity, it is natural to expect the
difference between CW and CCW propagation. Explanations need to be
given on why the observed chirality is unintuitive and unexpected.
Typo:
Page 15, line 204 : belief -> believe
Author Response
We thank the referee for the evaluation of our manuscript and the
valuable feedback which we incorporated in the revised version of
the manuscript.
A concern of the referee is that the main results should be
presented more clearly and their importance should be
highlighted. At the same time the referee relates the presented
results, i.e. the differences in the CW and CCW propagation
pattern, directly to the asymmetry of the cavity and therefore
finds the results expected and natural as he/she states "However,
given the asymmetry of the cavity, it is natural to expect the
difference between CW and CCW propagation.".
However, this is a misconception. The difference between CW and
CCW propagation which we discuss in the manuscript is not related
to an asymmetry in the cavity's boundary. This becomes
intuitively clear in Fig. 1b. Here, the CW propagating
counterpart of the CCW propagating red arrow is the blue arrow
which is not related to a mirror reflection but to a reversal of
the propagation direction. Thus even in a cavity with a symmetric
boundary such a difference pattern between CW and CCW propagation
occurs and organizes in an alternating sequence along the resonance
chain. For example, in the attached Figure the FPO
eigenstate difference for a symmetrically deformed microstar with
epsilon_1=epsilon_2=0.98 is shown.
We agree with the referee that this should have been communicated
more clearly. Therefore we have changed several paragraphs in
Sec. 1, 2, and 4. We have also added references in order to
distinguish our work from previous studies on asymmetric
backscattering which leads to an overall imbalance between CW and
CCW propagation.
With the simple quantum mechanical model system in the appendix
we show that the effect can be generalized apart form the context
of microcavities. It occurs in systems with a loss gradient
across a resonance chain. In microcavities however in can be
studied intuitively as forward and backward time evolution
manifests in "naturally" in CW and CCW propagation. We now say
this more clearly in the conclusion.
Some typos have been fixed in the manuscript. Figures 5 and 10
have been replaced due to a small correction.

Round 2
Reviewer 2 Report
I confirmed that the manuscript has been sufficiently improved by the authors' revision. Namely, in response to my comments, they added paragraphs to explain the novelty and importance of their results. Now I believe that the paper is acceptable for publication in Entropy.